

# Threshold-like associations as a function of disturbance

Kimmo Sorjonen[1], Michael Ingre[1,2,3] and Bo Melin[1]

[1] Department of Clinical Neuroscience, Karolinska Institutet, Stockholm, Sweden
[2] Department of Psychology, Stockholm University, Stockholm, Sweden
[3] Institute for Globally Distributed Open Research and Education (IGDORE), Stockholm, Sweden

## ABSTRACT

According to the intelligence-creativity threshold hypothesis, there should be a positive association between intelligence and creative potential up to a certain point, the threshold, after which a further increase in intelligence should have no association with creativity. In the present simulation study, the measured intelligence and creativity of virtual subjects were affected by their true abilities as well as a disturbance factor that varied in magnitude between subjects. The results indicate that the hypothesized threshold-like association could be due to some disturbing factor, for example, low motivation, illness, or linguistic confusion, that varies between individuals and that affects both measured intelligence and measured creativity, especially if the actual association between intelligence and creativity is weak. This, together with previous negative findings, calls the validity of the intelligence-creativity threshold hypothesis into question.

## INTRODUCTION

According to so called threshold hypotheses, the association between a predictor and an outcome is assumed to look different below and above a specific value, the threshold, on the predictor or alternatively on a third variable. It has, for example, been suggested that: (1) A certain minimum level of dominant language proficiency should be achieved before bilingualism can have a positive cognitive effect (*Cummins, 1976*, *1979*); (2) The economic growth in societies brings about an increase in the citizens' economic welfare (a measure combining income inequalities, social, and environmental factors), but only up to a point beyond which a further increase in economic growth may result in a deterioration of the economic welfare (*Max-Neef, 1995*); (3) A large enough difference in the quality of the territory held by an already mated and a non-mated potential partner may result in a fitness advantage of the bigamous compared to the monogamous choice (*Orians, 1969*).

According to the intelligence-creativity threshold hypothesis, high intelligence is necessary, but not sufficient, for high creativity (*Guilford, 1967*). This has led to the prediction that intelligence should have a positive influence on creative potential but only up to a certain level, the threshold is often set at IQ 120 (SD = 15), after which a further increase in intelligence should no longer influence creative potential (*Jauk et al., 2013*). This prediction has received some empirical support (*Cho et al., 2010*; *Fuchs-Beauchamp,*

Corresponding author
Kimmo Sorjonen,
kimmo.sorjonen@ki.se

*Karnes & Johnson, 1993*; *Jauk et al., 2013*; *Schubert, 1973*; *Shi et al., 2017*) although quite a few studies have failed to demonstrate such a threshold effect (*Kim, 2005*; *Preckel, Holling & Wiese, 2006*; *Runco & Albert, 1986*; *Sligh, Conners & Roskos-Ewoldsen, 2005*).

As examples of used measures of creativity, *Jauk et al. (2013)* measured creative potential with: (1) Three alternate uses tasks, where participants were asked to come up with as many novel and uncommon uses as possible for a can, a hairdryer, and a knife; (2) Three instances tasks, where participants were asked to come up with as many novel and uncommon solutions as possible to the problems "What can be elastic?", "What can make noise?", "What could one use for quicker locomotion?" The originality of the responses was rated by four students and both number of ideas (= ideational fluency) and originality were used as measures of creative potential. Creative achievement was measured with the Inventory of Creative Activities and Achievements, where respondents rate themselves, on a scale from 0 (= "I have never been engaged in this domain") to 10 (= "I have already sold some of my work in this domain"), across the eight domains arts and crafts, creative cooking, literature, music, performing arts, science and engineering, sports, and visual arts. *Jauk et al. (2013)* found a threshold-like association between intelligence and creative potential but not between intelligence and creative achievement.

When reading the articles about the intelligence-creativity threshold hypothesis mentioned above, for example noticing a few subjects with very low measured IQ (≈60) and creative potential in the plots in *Jauk et al. (2013)*, we started to suspect that this threshold-like association between intelligence and creative potential, if it exists, could, at least to some degree, be due to some disturbing factor, for example low motivation, illness, or linguistic confusion, experienced by some study participants. Of course, in a random sample of 300 individuals from the general population you would expect to get a few individuals with intellectual disability (IQ < 70), but we think that it is quite surprising to see this among subjects who have been recruited, as was the case in *Jauk et al. (2013)* through a local newspaper and the university's mailing list. Imagine, for example, a few friends who read about a study looking for participants who will get paid. They welcome the opportunity to make some extra money and respond. When filling out the questionnaires measuring IQ and creativity they do not put much effort into it, because they are doing this just for the money. When the researchers plot measured IQ and creativity against each other, these friends show up in the lower left corner. The objective of the present simulation study was to investigate the possibility that observed threshold-like associations might be due to the influence of disturbance.

## METHOD

### Simulation

Using R 3.5.0 statistical software (*R Development Core Team, 2018*) a dataset was simulated through the following steps (Fig. 1, both script and dataset available from https://osf.io/7b54w/): (1) Either 100, 400, 1,600, or 6,400 (i.e., four quadruplings) virtual subjects were allocated a true IQ score from a random normal distribution ($M = 100$, $SD = 15$); (2) The virtual subjects were allocated a true creativity score from a random normal distribution ($M = 100$, $SD = 15$) with a defined population correlation, drawn from
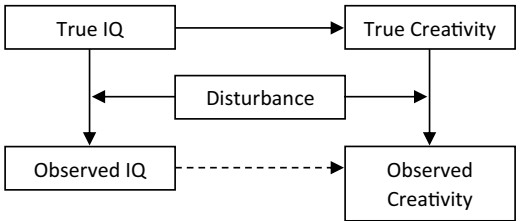

**Figure 1 Steps in the simulation and analysis.** Illustration of the steps in the simulation (solid lines) and analyzed effect (dashed line). 

random uniform distribution between 0 and 1, with the true IQ score; (3) The virtual subjects were allocated a disturbance score from a random beta distribution that was either negatively ($\alpha = 9$, $\beta = 1$) or positively ($\alpha = 1$, $\beta = 9$) skewed or approximately normally distributed ($\alpha = 9$, $\beta = 9$). These disturbance scores varied between 0 and 1, with a low value indicating a high degree of disturbance; (4) The virtual subjects were allocated observed IQ and creativity scores by multiplying their true IQ and creativity scores with their disturbance score, that is, the observed scores varied between zero and the true score. The present simulation did not assess the effect of measurement error other than the influence of the defined disturbance variable.

## Defining verification

The steps above were carried out 500 times for each of the 12 combinations of sample size and skewness of the disturbance variable. In these 6,000 simulations, and following *Jauk et al. (2013)*, the result was seen to support the threshold hypothesis if, and only if: (1) A significant breakpoint in the slope of the regression line was identified through segmented regression analysis (*Muggeo, 2008*); (2) The correlation between observed IQ and creativity was positive and significant below the breakpoint; (3) The correlation between observed IQ and creativity was significantly larger below compared to above the breakpoint. It should be noted that the threshold was not pre-defined, but rather empirically identified through segmented regression, again following the procedure by *Jauk et al. (2013)*. Also in accordance with Jauk et al. and others, we did not demand for the correlation above the breakpoint to be non-significant. One of the reasons why this custom has evolved in threshold research might be to avoid encouraging researchers to limit the amount of data they collect, as a large sample would increase the likelihood for a small correlation above the threshold to become statistically significant.

## Analyses

Analyses were carried out employing the segmented (version 0.5–3.0, *Muggeo, 2008*) and moments (version 0.14, *Komsta & Novomestky, 2015*) packages in R. In segmented regression, the predicted outcome, $E|Y|$, is given by *Muggeo (2008)*:

$$E|Y| = \beta_0 + \beta_1 z_i + \beta_2 (z_i - \psi) \times I(z_i > \psi) \tag{1}$$

In this equation, $\beta_0$ = predicted value on the outcome if the value on the predictor equals zero, that is, the intercept; $\beta_1$ = predicted increase in the outcome for an increase in the

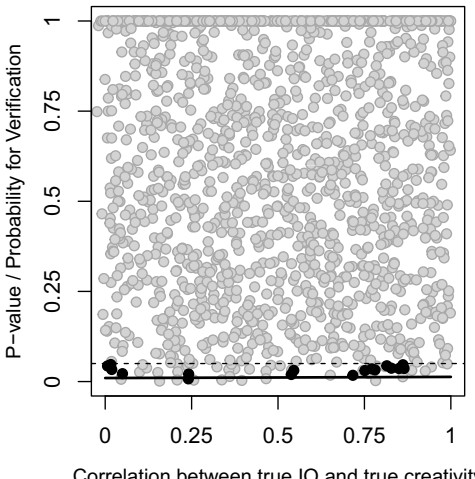

**Figure 2** **Association with no effect of disturbance.** *P*-values for the breakpoint according to segmented regression analysis (dots, black for results that support the threshold hypothesis and gray for results that do not) and probability for a result supporting the threshold hypothesis (solid line) as a function of the correlation between true IQ and true creativity. The dashed line indicates *P* = 0.05. 1,500 simulations with *N* = 6,400 in each.              

predictor by one, that is, the slope, given that the value on the predictor is below the tested breakpoint; $z_i$ = the value on the predictor of individual $i$; $\beta_2$ = difference in the slope above and below the tested breakpoint; $\psi$ = the tested breakpoint; $I$ = an indicator that equals one if the statement, that is, $z_i > \psi$, is true and zero otherwise. When conducting segmented regression with the segmented package in R, points along the continuum of the predictor are tested and the point resulting in the largest absolute value of $\beta_2$, with the additional requirement that the fitted lines should join at the breakpoint, is identified and if $\beta_2$ differs significantly from zero a significant breakpoint has been found (*Muggeo, 2008*).

For each of the 12 combinations of sample size and skewness of the disturbance variable, logistic regression was used to predict the probability for a result in accordance with the threshold hypothesis from the correlation between true IQ and true creativity.

## RESULTS

If disturbance was not allowed to influence the association between true IQ/creativity and observed IQ/creativity, that is, observed scores were identical to true scores, the predicted probability for a result supporting the threshold hypothesis was lower than the 5% nominal type I error rate (Fig. 2). However, if disturbance was allowed to influence the association between true IQ/creativity and observed IQ/creativity, a substantial probability to get a result in accordance with the threshold hypothesis could be observed, especially with a large sample size and a weak correlation between true IQ and true creativity (Fig. 3).

The effect of disturbance is illustrated in Fig. 4, with no correlation between true IQ (*M* = 120, SD = 15) and true creativity (*r* = −0.038, *P* = 0.453, Fig. 4A). A segmented regression analysis of the data in Fig. 4A indicated a breakpoint at true IQ = 163, but the breakpoint was not significant, that is, there was no significant difference in the slope of the regression line below and above the breakpoint (*P* = 0.167), and the association is not

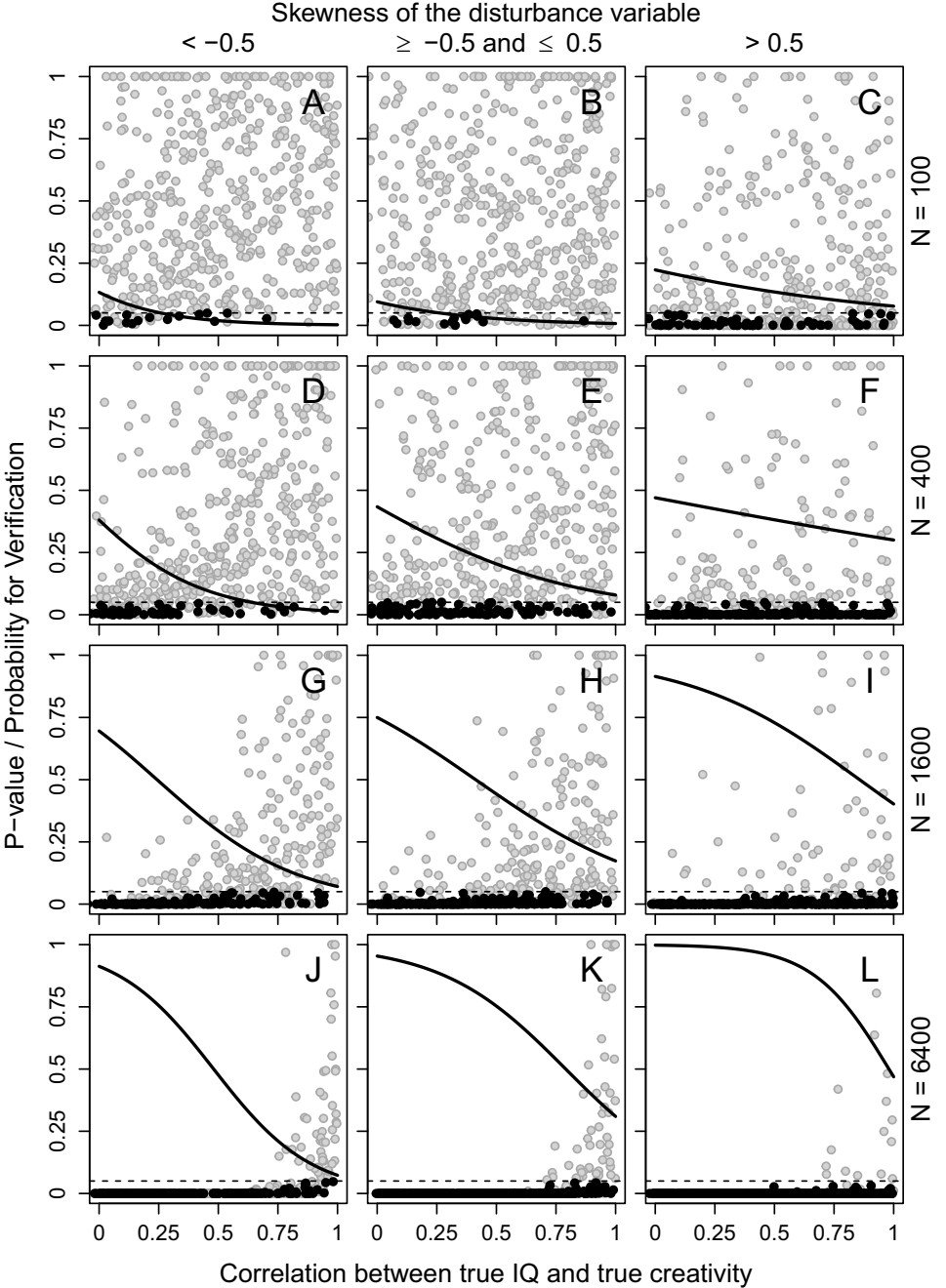

**Figure 3 Association with an effect of disturbance.** *P*-values for the breakpoint according to segmented regression analysis (dots, black for results that support the threshold hypothesis and gray for results that do not) and probability for a result supporting the threshold hypothesis (solid line) as a function of the correlation between true IQ and true creativity, separately for sample sizes 100 (A–C), 400 (D–F), 1600 (G–I), and 6400 (J–L), and for a negatively skewed (A, D, G, J), an approximately normally distributed (B, E, H, K), and a positively skewed (C, F, I, L) disturbance variable. The dashed line indicates *P* = 0.05. There are approximately 500 simulations in each panel.

in accordance with the threshold hypothesis. However, if we assume variation in the degree of disturbance experienced by the individuals, with some experiencing very high levels (indicated by a low value in Fig. 4B, the values were drawn from a random beta distribution
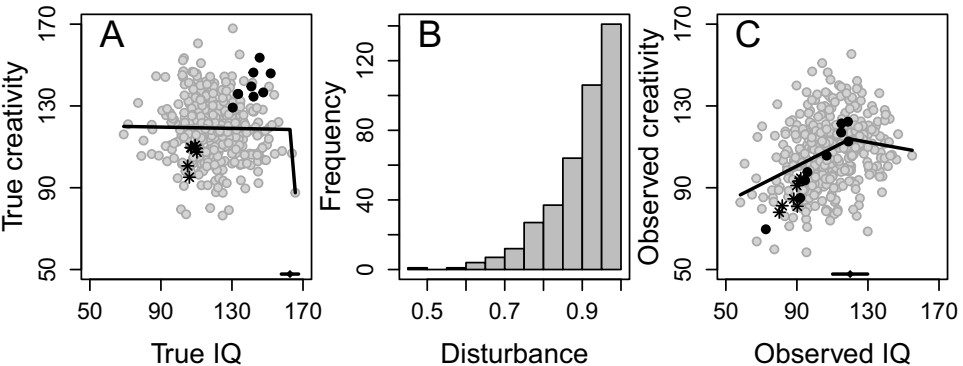

**Figure 4 Example of the influence of disturbance.** When two non-associated variables (e.g., intelligence and creativity) (A), are both multiplied with the same attenuating variable (e.g., motivation) that varies in magnitude between individuals (B), with low scores indicating high degree of disturbance (values have been drawn from a random beta distribution with $\alpha = 9$ and $\beta = 1$), we get two variables with a threshold-like association (C). Black dots = persons with high true IQ, high true creativity, and high degree of disturbance; Snowflakes = persons with low true IQ, low true creativity, and high degree of disturbance. $N = 400$. The line with a small dot at the bottom in (A) and (C) indicates the empirically calculated breakpoint with a 95% CI.

with $\alpha = 9$ and $\beta = 1$), and multiply the individuals' true IQ and creativity scores with their experienced degree of disturbance, we can observe a drift toward the lower left corner (Fig. 4C). Now we have an association that could be seen to support the threshold hypothesis, with a general correlation between observed IQ and observed creativity ($r = 0.306$, $P < 0.001$), a significant breakpoint at observed IQ = 120 ($P = 0.020$), and a significant positive correlation below the breakpoint ($r = 0.324$, $P < 0.001$) that is significantly stronger ($Z = 3.633$, $P < 0.001$) than the correlation above the breakpoint ($r = -0.093$, $P = 0.363$).

It is probably the individuals experiencing the highest degree of disturbance (i.e., those with the lowest score on the disturbance variable), and consequently the largest south-western drift, that are responsible for this phenomenon. If they are blessed with high true IQ and creativity (black dots in Figs. 4A and 4C), their drift moves them into the crowded center, thereby decreasing the correlation at the high end of the IQ scale. However, if they suffer from low true IQ and creativity (snowflakes in Figs. 4A and 4C), their drift even further south-west will strengthen the correlation at the low end of the IQ scale. As observed IQ, observed creativity, and residuals are quite nicely normally distributed (Figs. 5A–5C) and because there is no association between predicted degree of creativity and residuals from the segmented analysis (Fig. 5D), with a mean residual very close to zero (the horizontal line in Fig. 5D) for every level of predicted creativity, model diagnostics indicate a valid finding.

## DISCUSSION

The present simulation indicates that if subjects' observed scores on two variables that are not too strongly correlated, for example, IQ and creativity, are simultaneously affected by some disturbing factor, for example, low motivation, illness, or linguistic confusion, that varies in magnitude between subjects, there is a good chance of observing a threshold-like
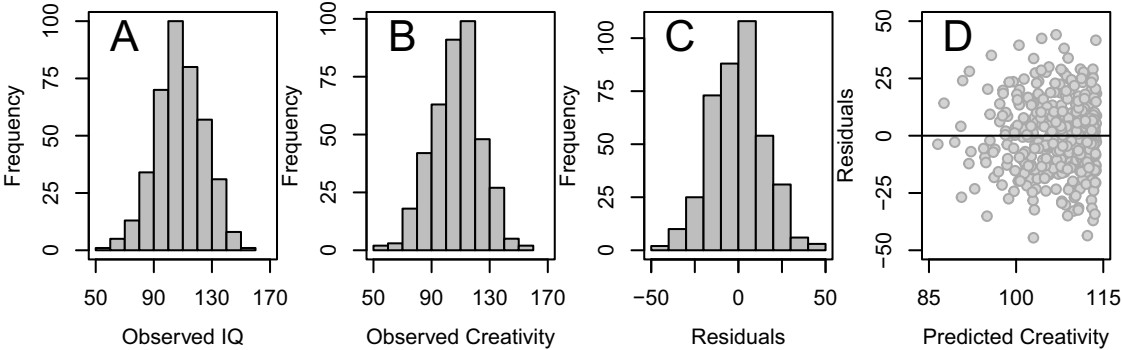

**Figure 5 Diagnostics of segmented analysis.** The frequency distribution of observed IQ (A), observed creativity (B), and residuals (= difference between observed and predicted creativity according to the segmented analysis) (C), as well as the association between predicted creativity and residuals (D).

association between the two variables, at least if the analysis is not crippled by low power due to a small sample size.

We propose that such a possible influence of disturbance is a threat mainly against the validity of studies investigating threshold-like associations between abilities, for example, IQ and creativity, that are operationalized as measured performances. The validity of such measures is not guaranteed, as lack of disturbance, together with high ability, is needed for a high performance, while presence of disturbance is enough for a low performance. Therefore, we could probably assume that the validity is positively correlated with measured performance and, consequently, tends to be lower among those with low scores. To paraphrase the threshold hypothesis: a high true intelligence/creativity is necessary, but not sufficient, for a high measured intelligence/creativity.

A consequence of the above thoughts might be a prediction that performances, generally, tend to have threshold-like associations with each other, as long as they are not perfect indicators of true abilities, the true abilities are not too strongly correlated with each other, and they are simultaneously influenced, at least to some degree, by the same attenuating factors. For example, decathletes' performances on pole vault and shot put might exhibit a threshold-like association due to some of the athletes' mild illness, sore shoulder, or low motivation. *Spearman's (1927)* "law of diminishing returns" reflects the observation that IQ test scores tend to be more highly correlated among those with low, compared to those with high, ability (*Deary et al., 1996*). Due to its similarity with the threshold hypothesis, both suggesting a stronger association at the low end of the IQ scale, it is conceivable that both are due to the same cognitive mechanisms, for example, higher degree of differentiation of abilities at higher levels. However, yet another possibility is that findings in accordance with Spearman's law are also, at least to some degree, due to an influencing effect of disturbance.

As mentioned in the introduction, the threshold-like association between intelligence and creativity has revealed itself in some studies but not in others. One, maybe the most likely, explanation of this could be simple random variation. In the light of the present findings, another possibility is that studies who have failed to detect this threshold have been less affected by disturbing factors with a simultaneous influence on measured

intelligence and creativity. We can also note that the threshold-effect seems more elusive when creativity is measured as actual achievement rather than as potential (*Jauk et al., 2013*). This difference seemingly fits with the present findings, as high scores on tests of creative potential, for example trying to come up with as many and as original alternate uses for a hairdryer as possible, are probably more reliant on the absence of disturbing factors, for example, low motivation, than are high scores on tests of creative achievement, for example answering questions if you have produced something in the areas of literature or creative cooking.

Recently, a few studies have employed a newly developed method called necessary condition analysis (NCA, *Dul, 2016*) and claim to have found support for the notion that a certain minimum level of intelligence is necessary for a high level of creativity (*Karwowski et al., 2016*, *2017*; *Shi et al., 2017*). However, as a negatively skewed predictor (e.g., measured intelligence) and a positively skewed outcome (e.g., measured creativity) seems to be enough for NCA to indicate a strong effect (*Sorjonen, Wikström Alex & Melin, 2017*), the validity of these claims can also be called into question. Another way to evaluate the association between intelligence and creativity could be to apply both non-linear and segmented linear models to collected data and to compare the fit of these. This would give an indication if the association is best described by a gradual (e.g., quadratic) or a sharp (segmented linear) change in the slope. However, it is possible that in either case the observed association could still be due to an influence of disturbance.

This study has some limitations. For example, for the disturbance variable we used beta distributions (because these are bounded by 0 and 1, which seems logical for an attenuating variable) that were either "substantially" positively or negatively skewed or approximately normally distributed. Although the main result of this study did not depend on which of the three distributions was used, there is of course a myriad of other possible distributions that could have been employed, for example with extreme negative skewness ($\alpha = 100$, $\beta = 1$), uniform ($\alpha = 1$, $\beta = 1$), or bimodal ($\alpha = 0.5$, $\beta = 0.5$) distributions[1]. On balance, we believe that a negatively skewed disturbance variable similar to the one in Fig. 4B should be quite realistic in many situations. It indicates that most subjects would get an observed IQ and creativity score close to their true value (disturbance close to 1) while a few receive a score well below their true value (only half of it in the most extreme case). Another limitation is that in each simulation we used only one disturbance variable and this was common both for intelligence and for creativity. In reality, there would probably be several different disturbance variables in play, some of them common and some unique only for intelligence or for creativity.

Somebody might be tempted to argue that because this simulation has limitations, for example that we do not know the exact distribution of the disturbance variable, its results are not trustworthy and, therefore, (1) it does not disprove the intelligence-creativity threshold hypothesis with absolute certainty, and (2) the intelligence-creativity threshold hypothesis must, consequently, be considered correct. We admit that we cannot be sure that an effect of disturbance is the sole or even one of the explanations of the intelligence-creativity threshold association demonstrated in various studies. However, we, as well as many others (*Popper, 1959*), would argue that "X has not been proven false with

[1] If somebody happens to know the true distribution of the disturbance variable, they are welcome to implement this into the available script, or to inform us so that we can do it.
absolute certainty" does not necessarily imply that "X is correct." We believe that the present simulation points at the possibility that at least some of the findings verifying the threshold hypothesis could be due to an influence of disturbance. It can also be noted that quite a few published studies, including the meta-analysis by *Kim (2005)*, found no support for the threshold hypothesis. This, together with the high estimated rate of publication bias in psychological research (*Ingre & Nilsonne, 2018*), and the results of the present simulation, could be seen to indicate that there is, for the moment, more speaking against than in favor of a true threshold-like association between intelligence and creativity.

### Funding
The authors received no funding for this work.

### Competing Interests
The authors declare that they have no competing interests.

### Author Contributions
- Kimmo Sorjonen conceived and designed the experiments, performed the experiments, analyzed the data, prepared figures and/or tables, authored or reviewed drafts of the paper, approved the final draft.
- Michael Ingre conceived and designed the experiments, authored or reviewed drafts of the paper, approved the final draft.
- Bo Melin conceived and designed the experiments, authored or reviewed drafts of the paper, approved the final draft.

### Data Availability
Data and script are available at Open Science Framework: Sorjonen, Kimmo. 2019. "Threshold-like Associations as a Function of Disturbance." OSF. June 14. https://osf.io/7b54w/.

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
