# Peer review of "Threshold-like associations as a function of disturbance"

_PeerJ, doi:10.7717/peerj.7891_

## Round 0.1 · original submission · Major Revisions

I found your manuscript to be an interesting demonstration of how a simple association can be distorted to appear more complex through measurement error. The reviewers have made some very helpful suggestions and I’ll ask you to address each of their questions/points, with revisions or a rebuttal provided for each, and the same for my questions/points below. Overall, I would like to see the rather short discussion bolstered, and some of the reviewers’ and my suggestions should help with that.

The process here will generate graduate turning of the line showing the association (as can be shown using locally weighted splines—note that I’ve performed similar but not identical simulations when thinking about your manuscript) rather than more precise segmentation at a particular breakpoint (as with linear splines and as modelled by segmented linear regression). I wonder if this could be discussed alongside some results from the literature. Do published graphs and/or data sets support your theory and do they suggest this gradual turning rather than a sharp cancelling/changing of the association? I think showing instances of the data from the literature would help provide more context to the reader for your intriguing suggestion.

Related to this, the threshold hypothesis for intelligence-creativity (Lines 3–4 and 30–32, as distinct to the general idea of a threshold hypothesis on 17–18) claims no further association between IQ and creativity beyond some level of IQ, but you are treating a significantly lower slope as evidence for this (lines 81–82) which is consistent with the more general definition of a threshold effect. Can you explain why you have not required a lack of evidence for a correlation above the threshold? You could include a version of Figure 3 with the more demanding probability graphed (segmentation with a positive association below and no association above the breakpoint).

I wonder if you have considered separate (but correlated to various degrees) disturbance factors for IQ and creativity. This seems more plausible to me than a single shared disturbance factor, and although it would add another dimension to the simulations, it would, I think, increase the realism of the simulations.

I also wonder how the disturbance being distributed with a beta distribution can be justified (Reviewer #1 raises a similar point). Why not uniformly distributed, for example, or more peaked (e.g. beta(99,1))? There didn’t seem to be any justification for this, and I found the positively skewed beta in particular to be rather implausible in terms of realistic effects (I guess this could apply under conditions of extreme tiredness though with a small number of participants resilient to the effect).

It seems to me that inducing this threshold effect as described here also affects the variability of the data and induces heteroscedasticity (which Figure 4’s rightmost panel would cause me to wonder about), which should then be picked up in model diagnostics, and might encourage a log-transformation (or some other approach). Since this will also (partially or fully) linearize concave down associations, such as shown in Figure 4, is the problem here to some extent one of inadequate model diagnostics?

·

Basic reporting

no comment

Experimental design

The research context is based in the educational and social sciences. It is unclear how it falls within the bounds of Biological Sciences, Environmental Sciences, Medical Sciences, and Health Sciences. However, the methodology is rigorous and described sufficiently, and the research question is relevant and meaningful. There is a potential for the methodology to be useful within the scope of the journal.

Validity of the findings

See general comments.

Additional comments

The current study presents simulations of possible relations between creativity and intelligence to test conditions under which the threshold theory would or would not be supported. The argument is that something other than the assumed necessity of IQ for creativity could produce a threshold-like outcome, especially when sample sizes are large (e.g., N = 400+) and the true correlation is low (e.g., < .25). The other variable, called a disturbance, is due to individual differences other than in IQ.

I think this is a very interesting study that highlights that threshold hypotheses of any kind can be produced under circumstances other than the assumption that one variable is necessary but not sufficient for a second variable. In the case of creativity, IQ is considered as necessary for creativity but not sufficient: that is, as IQ increases, creativity also increases to a certain point (threshold) of IQ, after which creativity does not increase. However, I have some concerns that it is not tied enough to the research on creativity. My concerns revolve around the question, what might the disturbance be? I highlight these below.

1. The authors mention a few possibilities in passing (e.g., low motivation, illness, linguistic confusion). But these mentions are too general for an enhanced understanding of creativity. How might these possibilities be related to creativity specifically? Or IQ? Some concrete examples of a variable that would affect both creativity and IQ would be helpful.

2. Why choose skewed distributions for the disturbing factor? Again, some concrete examples of what these might represent would be helpful.

3. Why would we find the threshold with some creativity tasks but not others, as some previous researchers have found? How does this relate to disturbances? More generally, the simulations, though impressive, do not explain why some studies find a threshold-like relation and some don’t. Is the claim that in some studies there are disturbances but not in others?

4. One of the hallmarks of a threshold theory is that the relation between two variables in qualitatively different above the threshold than below it. If I’m understanding correctly, the authors claim that this is illusory – a fluke of the individual differences in disturbance. What does this mean for creativity theory more generally?

5. It seems remarkably easy to find evidence for the threshold theory, especially when the correlation between IQ and creativity is low (i.e., < .25), no matter the sample size or skewness of the disturbance –the probability is above .05 in all these cases. Why don’t we find it more often in our creativity research?

6. Other minor comments.
(a) Something is missing from the first sentence of the introduction.
(b) It would be easier to understand the extent to which the simulations present evidence of a threshold if the black dots in Figure 3 were in front of the gray dots.
(c) When there is no correlation between IQ and creativity, as shown in the left side of Figure 4, there is essentially no threshold, even though there may be statistically. No sample was above IQ = 165.

·

Basic reporting

The manuscript is well-written, concise, easy to comprehend, and makes appropriate reference to other works. The figures support the text in an excellent manner. The data are accessible.

Experimental design

The design of this simulation study is – as far as I can tell as a psychologist, not statistician – sound and appropriate to investigate the proposed hypothesis. The methods are described transparently and in sufficient detail to understand the simulation results. However, for a more comprehensive picture, I wonder whether a baseline simulation for true positives (sensitivity analysis) might be useful. The probabilities of detecting false and true positive could be compared, then.

Validity of the findings

Alternative models explaining the emergence of threshold-like distributions should be considered. The current model focuses on disturbance as a variable which attenuates the true scores. However, this is only one of many possible theoretical assumptions, and others would be plausible as well. Particularly, the threshold effect is consistent with the differentiation hypothesis (cognitive abilities are more differentiated, and thus less correlated, at higher levels), as proposed by Spearman (and confirmed in large samples by Deary).
I understand that the simulation is well suited to reproduce bivariate threshold-like distributions, and I very much appreciate the idea, but it should be acknowledged that there might be many other ways of reproducing such distributions. Whether attenuation of the true scores or different g-saturation of true scores, for example, are better suited to explain the observed associations would either require (a) different simulation models being compared against each other and/or (b) empirical research (varying disturbing or motivating factors, for instance). In my view, this study provides a great starting point for this.

Additional comments

First of all, I apologize for my delay in this revision process!
The manuscript under review reports on a simulation study of disturbing factors in threshold-like associations detected with segmented regression. The hypothesis that disturbing factors such as low motivation could generate such associations among two correlated variables is confirmed.
I am very happy to see that methodological research is devoted to the use of segmented regression in psychology. I find the proposed idea and simulation very interesting. My main point is that the interpretation should more explicitly acknowledge that different models could produce such distributions.

---

## Round 0.2 · Minor Revisions

Both reviewers are happy with your revisions and responses. Please note Reviewer #2’s suggestion around the distribution of IQ scores, which you may wish to incorporate into your manuscript (see my comment below). There are a few smaller issues that still need addressing, which I’ll list below, and look forward to seeing your revised manuscript in due course. I don’t expect that any of these will require too much effort.

Lines 34–36: As reviewer #1 pointed out previously, this sentence needs attention. One option would be to add “or” between “predictor” and “alternatively” on Line 36.

Line 48: While the difference is trivial this close to 100 even when using the older versions of the Stanford-Binet, and this is no longer an issue for newer versions of this, it might be useful to clarify whether this is Wechsler or Stanford-Binet IQ or perhaps even better just to provide the standard deviation assumed (presumably 15) here.

Line 49: This doesn’t quite make sense (“influence an increase”). Perhaps “longer influence creative potential” or “longer increase creative potential”.

Line 70: See Reviewer #2’s comment. As they point out, you would expect 1 or 2 such individuals in a sample of 300 (this is where the number expected does depend on whether Wechsler/current Stanford-Binet or an older version of the Stanford-Binet is being used). Whether you have noted more than these and the sampling frames used might be useful to explain here for the reader’s benefit.

Line 87: To save the reader needing to check the code, perhaps you could mention with a mean of 100 and a standard deviation of 15 here. Not that this is important to the interpretation of the results, but it is a detail some readers will want to know. The same applies to Line 88 for creativity.

Lines 93–95: This does assume that observed scores cannot exceed true scores, which will not necessarily be the case due to guessing, for example. You could clarify that the lack of measurement error here, given a “person’s” true score and their disturbance factor, is intentional.

Lines 107–110: This is awkwardly worded. Do you mean, “One of the reasons why this custom has evolved in threshold research might be to avoid encouraging researchers to limit the amount of data they collect, as a large sample would increase the likelihood for a small correlation above the threshold to become statistically significant.”?

Lines 113 and 114: Please give the version numbers for the package to help with reproducibility (default values can change between versions).

Line 136: Perhaps “…was lower than the 5% nominal type I error rate…” for clarity here.

Lines 141–142: It might be worth mentioning the mean of 120 (and standard deviation of 15) used here in case the reader wonders about the centre of mass for IQ being clearly greater than 100.

Line 146: It would be useful to clarify that this is from a beta(9,1) distribution here.

Line 164: I think this could be improved by referring to validity (in the internal sense) rather than reliability (i.e. repeatability) and by making observations about the model diagnostics rather than speculating on what researchers (a small number of who will be (bio)statisticians) might think.

Lines 213 and 215: A quadratic association is of course only one of many options here and I would suggest “non-linear” as a more general description of the possibilities on the first line at least (or perhaps for both lines). If you changed Line 213 to “non-linear”, you could prepend “for example” to “quadratic” on Line 215.

Line 218: I wouldn’t generally expect to see “lion share” in an academic work and note that the idiom is “lion’s share” (the share that belongs to the lion) and refers to having the major share of something. I’m not sure that this work has appreciably more limitations than other research, so I’m not sure that you do indeed have such a large share of limitations. Perhaps delete “lion share of” from here.

Line 222: Perhaps “…there is of course a myriad of…” (a myriad, as a noun, already refers to many things). See also Lines 238 and 241 below.

Line 225: I suggest deleting “If we have to guess,” as this is rather non-scholarly (to my ear). An alternative would be “On balance, we believe…” if you wish to convey a degree of uncertainty here.

Lines 229–230: This is non-traditional, which is not to say that it shouldn’t be included, but I wonder if a footnote would not be more appropriate for this request.

Lines 231–237: I think that this paragraph could be simplified by using fewer examples and still get its point across as it does go through more possibilities than I would expect to see identified: “Another limitation is that in each simulation we used only one disturbance variable and this was common both for intelligence and for creativity. In reality, there would probably be several different disturbance variables in play, some of them common and some unique only for intelligence or for creativity, some with distribution X and others with distribution Y or Z, some with a positive and others with a negative correlation with each other and some unrelated, some with a positive correlation with intelligence/creativity, i.e. experienced mainly by those with high ability, and some with a negative correlation while others unrelated with true ability, etc. So, there are, again, myriads of different possibilities to explore further.” Perhaps this could stop after “or for creativity” on Line 234?

Figure 1: I understand why and I doubt it will cause any real confusion, but could you fit “Observed” into the two lower boxes, perhaps by changing the font and/or box size?

Figure 2: A similar point about abbreviations. Could the words be included in full, perhaps by using a smaller font for the axes labels and titles? (As you do in Figure 3.)

Figure 3: Same point for “Skew.Dist.” Could you also use non-strict less than inequality symbols (≤) rather than compound symbols (<=).

Figure 4: This panel doesn’t show the distribution of the model residuals, which would be a key aspect in judging the model as appearing to satisfy the model assumptions. Note that the usual purposes of plotting residuals against fitted values (panel F) are a) to check for non-linearities and b) to check for evidence against homoscedasticity, and the latter did not appear to be addressed in the text. Positive skew in the model residuals and/or evidence of increasing variance with greater fitted values would support investigating a log-transformation, which, as I noted previously, would, to some extent, linearise the association in panel C. I’m not suggesting that this is an issue here, but for the reader to be confident of that, these diagnostics should be addressed. This could also be done by descriptive explanations in the text rather than adding graphs here. Note that the titling and labelling of y- and x-axes is inconsistent here and all graphs should have both identified and include tick marks. Finally, it would be helpful to note that panel B shows data from a beta(9,1) distribution.

·

Basic reporting

Basic reporting is excellent.

Experimental design

The experimental design is sound.

Validity of the findings

The findings are valid.

Additional comments

The authors addressed all my concerns in their revision. I will say, in relation to a comment from the authors, that I have difficultly finding evidence for a threshold in my lab.

·

Basic reporting

No comment

Experimental design

No comment

Validity of the findings

No comment

Additional comments

The authors addressed my points from the previous revision - I am happy with the present form of the manuscript. Though the authors did not follow my suggestion to include simulations targeting the sensitivity of the method, I understand this goes beyond the scope of the present work.

As a very minor point, I would like to note that observing a few individuals with IQ ~ 60 in a sample of N ~ 300 is actually in line with the expected frequency based on a normal IQ distribution. The authors mention this as evidence for disturbance at different occasions, but it could also be seen as a sign of a valid sampling that includes individuals from all IQ levels (which is usually not the case in student samples). Maybe the authors could consider this perspective in their final manuscript.

I think this work provides important methodological considerations for creativity researchers, and more generally, for those investigating threshold-like associations, and will stimulate future research in a constructive way!

---

## Round 0.3 · accepted · Accept

Thank you for your revisions. I am very happy to accept your manuscript and look forward to it generating some stimulating discussions.

I will note a very small possible typo on Line 187: “hypothesis: A high true intelligence/creativity is necessary…” where the “A” should perhaps be “a”. The comma on Lines 246 in “association, demonstrated” seems unnecessary also. These can easily be addressed in the proofing stage.